# CONFESSION NETWORKS: BOOSTING ACCURACY AND IMPROVING CONFIDENCE IN CLASSIFICATION

## ABSTRACT

We present a novel method for measuring the confidence of neural networks in classification problems in this paper. There are existing statistical approaches to measure neural network confidence for classification. However, in this paper, we suggest a new loss function such that the neural network signals the amount of confidence it has for its prediction, independent of the prediction itself. The first objective of this article is to design an appropriate loss function to output a confidence measure along with classification scores for neural networks. A second goal is to examine whether such a loss function can improve network performance. There are many applications where a confidence measure is essential, including autonomous driving to ensure that the predictions relating to the area around the vehicle are correct or in important medical diagnostic decisions. We demonstrate that the proposed approach both improves prediction accuracy and also provides a valuable output for gauging the confidence of the prediction.

## 1 INTRODUCTION

Artificial Neural Networks are versatile and find application in tasks that involve regression as well as classification. Papadopoulos et al. (2001). This paper focuses on classification tasks, but our method could be used for other tasks with minor modifications. Classification tasks are typically difficult because large-scale labeled data is hard to come by with data collection being expensive and time-consuming, especially in specialized domains. Given both the lack of data, and as a general problem, network reliability could be improved by having a direct measurement of confidence in the prediction made by the network. In this paper, we concentrate on supervised classification problems. In this type of classification, training data have labels and class memberships are known. The assigned class is typically given the highest score at the output layer in considering all classes. It is not clear, however, to what degree this translates into a measure of confidence in the prediction. That is, for a given class and a given score, there is no natural way of determining the confidence of the classification from this value alone. Our contribution in this part is to design a network that confesses its confidence for the prediction it has alongside the classification prediction.

One motivation for this work is in decision-critical settings. For instance, in autonomous driving, there is a need to be sure about the prediction of entities in the surrounding area of the vehicle. In healthcare settings, we need to be sure about classifying disease correctly or have a means of deciding when to signal that an image should be looked at by an expert with human intervention involved.

There are some existing methods that attempt to measure the confidence of predictions made by neural networks. However, almost all prior work tries to use statistical methods to estimate the confidence of neural networks. Moreover, there are some articles by Zhao Xu et al. Xu et al. (2022) and Ramya Hebbalaguppe et al. Hebbalaguppe et al. (2022) that focus on increasing confidence. Jishnu Mukhoti et al. Mukhoti et al. (2020) designed a Focal loss to train Neural Networks to have more confidence and accuracy. Liu Ziyin et al. Liu et al. (2019) add new nodes to Neural Networks to increase confidence by employing portfolio theory. Differing from the techniques presented in these scholarly articles, we put forth an innovative approach to indicate the confidence level of neural networks when dealing with classification tasks. In this work, not only do we increase the accuracy of Neural Networks, but we also have an additional output, which is the confidence of the network. Unlike other works that aim to increase the confidence of NNs, we not only increase it but also have

the confidence as an output for each individual input which could be very valuable in real-world applications like autonomous driving. There are three central contributions to this paper:

1. Our proposal entails the introduction of a novel architectural framework designed for classification tasks. Within this framework, neural networks incorporate an additional element referred to as the "*confidence node*," which serves the purpose of directly forecasting confidence levels.

2. We designed a new loss function, which includes the new node, to compute the confidence of a network while simultaneously performing classification. Our loss function makes training occur in a manner that returns the confidence of a prediction as well as the class-wise predictions with both having an impact on the weights derived during back-propagation.

3. We provide an extensive assessment of the newly introduced loss function in contrast to conventional loss functions. This evaluation is conducted across CIFAR-10 and CIFAR-100 datasets, revealing notable enhancements in prediction accuracy when compared to simpler standard loss functions.

The rest of this article is structured as follows: Section 2 delves into established methods for assessing prediction confidence in classification. Section 3 outlines the approach and techniques used in the new architectural design and loss function. Section 4 encompasses the assessment and outcomes derived from our experiments, and finally, Section 5 encapsulates the conclusion and insights drawn, which in turn serve as inspiration for prospective research endeavors.

## 2 BACKGROUND

### 2.1 CONFIDENCE MEASUREMENTS

Here, we provide a concise overview summarizing various methods that have been employed to gauge the confidence levels of neural networks.

Khawaja et al. (2005) introduced a novel technique for computing the uncertainty of neural networks in their predictions Khawaja et al. (2005). Their research primarily emphasizes the substantial challenge posed by the representation and handling of uncertainty within Condition Based Management Systems in the context of fault prognosis. It underscores the significance of efficiently representing and reducing uncertainty boundaries as more data becomes accessible for the extended-term prediction of time to failure. The paper identifies prediction accuracy and precision as pivotal performance metrics for assessing prognostic algorithms, with the objective of minimizing the disparity between predicted and actual time to failure while narrowing the uncertainty bounds. To address these challenges, the paper introduces the Confidence Prediction Neural Network (CPNN) as an innovative approach. This approach operates under the assumption that neural network prediction errors adhere to a normal distribution, and confidence levels are computed based on measurements derived from this normal distribution. Furthermore, the review underscores the inherent complexity in predicting time to failure while grappling with uncertainty, particularly highlighting the limitations of traditional statistical models like ARIMA when computing prediction intervals. It also points out that the standard deviation of prediction errors, commonly used in such contexts, falls short in addressing multi-step prediction challenges. Overall, the literature review underscores the critical role of addressing uncertainty in fault prognosis and presents the proposed CPNN as a promising solution, supported by real-world examples and the incorporation of learning routines for uncertainty management.

Deep neural networks achieve impressive accuracy but often suffer from miscalibration and overconfidence. Techniques such as temperature scaling (TS) Ding et al. (2021) and label smoothing (LS) Müller et al. (2019) have been successful in improving model calibration in the past by applying scalar factors to smooth logits and hard labels, respectively. However, these methods may not be optimal for long-tailed datasets, where the model is overly confident for classes that have high-frequencies. Mobarakol Islam et al. Islam et al. (2021) focus on the proposal of class-distribution-aware LS (CDA-LS), and TS (CDATS), which incorporate class frequency information to enhance model calibration. The experimental results indicate that CDA-TS and CDA-LS techniques effectively enhance model calibration and accuracy on ImageNet-LT, CIFAR-100-LT, and Places-LT

datasets. While CDA-LS outperforms traditional LS techniques in most metrics, scalar LS techniques show better performance in certain uncertainty calibration error metrics when dealing with high-class imbalance. Additionally, CDA-TS does not hinder knowledge distillation and exhibits superior accuracy compared to baseline models and models trained using TS. Notably, CDA-TS mitigates calibration errors more effectively than TS when dealing with highly imbalanced data. Overall, this research contributes valuable insights into addressing model calibration challenges in deep learning scenarios.

Zhao Xu et al. Xu et al. (2022) focus on the measurement of uncertainty in GNNs specifically for node classification tasks. While most existing GNNs assume deterministic message passing among nodes Kipf & Welling (2016), this work addresses the presence of uncertainty in these messages and explores methods to propagate such uncertainty throughout the graph. In response to these challenges, the authors present a Bayesian Uncertainty Propagation (BUP) technique that embeds Graph Neural Networks (GNNs) within a Bayesian modeling framework. This novel approach addresses the issue of predictive uncertainty in node classification by considering both the Bayesian confidence associated with predictive probabilities and the uncertainty inherent in message passing. The BUP method introduces an innovative mechanism for propagating uncertainty, drawing inspiration from Gaussian models, and employs a customized loss function that prioritizes mitigating high uncertainty predictions during the learning process for node classification. The authors substantiate the effectiveness of the BUP approach in terms of enhancing prediction reliability and handling out-of-distribution (OOD) predictions. Furthermore, they conduct a comprehensive examination of the acquired uncertainty, including its correlation with graph topology and its impact on predictive uncertainty in OOD scenarios. The empirical results obtained from widely recognized benchmark datasets provide evidence that the proposed approach outperforms state-of-the-art (SOTA) techniques. The conducted tests also delve into the prediction uncertainty observed in out-of-distribution (OOD) scenarios and investigate the correlation between uncertainty and network architecture.

Liu Ziyin et al. Liu et al. (2019) propose a novel approach for selective classification, which involves supervised learning with a rejection option to achieve the best performance at a desired data coverage level. They transform the m-class classification problem into an (m+1)-class problem, incorporating an additional class representing 'disconfidence'. Inspired by portfolio theory, they introduce a loss function based on the doubling rate of gambling, enabling a balance between confident predictions and abstention. This end-to-end method allows for flexible decision-making during prediction and achieves competitive performance without requiring significant modifications to the model. Experimental results demonstrate the method's ability to identify uncertainty, achieve strong performance on datasets, and learn superior hidden representations. The authors also highlight potential applications in scientific research, interpretability, and improved model robustness, pointing to promising future research directions.

Dan Hendrycks et al. Hendrycks & Gimpel (2016) focus on two interrelated problems in machine learning: error and success prediction, and in- and out-of-distribution detection. The authors propose a straightforward baseline approach that utilizes probabilities obtained from softmax distributions to address these challenges. The baseline method exploits the observation that correctly classified examples typically exhibit higher maximum softmax probabilities compared to out-of-distribution examples. The proposed baseline approach leverages prediction probabilities derived from softmax distributions. The authors reveal that misclassified samples tend to have lower probabilities in prediction compared to correctly classified examples. This insight suggests that capturing prediction probability statistics of correctly classified or in-sample instances is usually adequate for erroneous or abnormal instance detection, even though individual prediction probabilities alone may be misleading. The effectiveness of this baseline is evaluated across several tasks in machine learning demonstrating its applicability and performance. This work contributes by establishing evaluation metrics for assessing the automatic detection of errors. They propose techniques to introduce out-of-distribution examples during testing, such as incorporating images from different datasets or realistically distorting inputs. However, the study also highlights cases where the baseline approach can be improved upon, emphasizing the need for further research in these relatively unexplored detection tasks.

## 2.2 LOSS FUNCTIONS

Hu et al. (2018) proposed an improved Cross-Entropy loss function to create the probability map from image to image with the use of a Convolutional Neural Networks (CNN) for Retinal vessel analysis Hu et al. (2018). Their proposed cross-entropy loss ignores fractional loss in an easy sample in order to learn hard samples. Their approach gave their CNN better accuracy than Regular CNNs with Mean Absolute Error (MAE). Zhang et al. (2018) propose using Categorical Cross Entropy (CCE), which is a generalization of MAE and Cross-Entropy (CE) Zhang & Sabuncu (2018). They claim that MAE works poorly with Deep Neural Networks (DNN). They showed that their proposed loss function is noise robust and could be easily applied to any deep neural network. Hadi Goldani et al. (2021) made use of Margin loss instead of cross-entropy to train their CNN to detect fake news on social media Goldani et al. (2021). They compared word embeddings with non-static embeddings, which helped them update word embeddings in the learning phase of their network. Eventually, this produced much better results than other loss functions on the same dataset.

Deep Neural Networks (DNNs) often suffer from overconfident mistakes. SOTA calibration techniques have emerged to improve the reliability of the prediction, however, they often fall short of adequately addressing the calibration of non-maximum probability classes and lack pixel-specific calibration. Consequently, these methods are not well-suited for tasks requiring label refinement and dense prediction. Ramya Hebbalaguppe et al. Hebbalaguppe et al. (2022) propose an innovative approach, presented in their recent work titled "Stitch," which intervenes during the training phase to directly produce calibrated DNN models. The proposed method introduces the MDCA auxiliary loss function, which significantly improves calibration in various scenarios, including image classification, segmentation tasks, domain shift, imbalanced data, and natural language classification. The approach provides a trainable and practical solution, rivaling post-hoc methods while eliminating the need for a hold-out set. The method's robustness is evident in maintaining calibration under dataset drift and imbalanced datasets.

Rosset et al. Rosset et al. (2003) present the margin maximization in classification models, emphasizing its importance in both theoretical and practical aspects. Margin maximization offers a clear interpretation of the networks being constructed, facilitating the analysis of generalization errors. The authors put forth a sufficient condition pertaining to regularized loss functions, these functions will converge toward margin-maximizing separators. This condition is applicable to well-known loss functions, including the hinge loss, the exponential loss employed in AdaBoost, and the logistic regression loss.
The primary outcome of this paper revolves around the establishment of the aforementioned sufficient condition for margin maximization. This condition reveals that widely used loss functions, such as 1-norm SVMs, L1-regularized logistic regression, and exponential boosting, all converge to the same non-regularized solution as the regularization term approaches zero.

Rusiecki Rusiecki (2019) proposes a simple solution to enhance the robustness of deep neural network training when confronted with noisy labels. Without modifying the network architecture or learning algorithm, they introduce a modified loss function which leads to enhanced generalization when trained in scenarios characterized by label noise.
Deep neural networks have garnered significant attention due to their capacity to glean high-level abstractions from extensive supervised datasets. Nonetheless, the performance of these data-driven models heavily relies on large, accurately annotated datasets. When the data annotations contain errors or label noise, the reliability of these models decreases. Nonetheless, creating well-annotated training data is often a laborious and costly process. Even when data labels are generated through automated algorithms or web search engines, they remain susceptible to the noise existing in the labels. This article addresses the issue of label noise by proposing a straightforward approach that enhances the training process without requiring any changes to the network architecture or learning algorithm.

Jishnu Mukhoti et al. Mukhoti et al. (2020) explore the use of focal loss as an alternative to the cross-entropy loss for training classification networks. Focal loss addresses the problem of miscalibration in deep neural networks by regularizing the KL divergence and increasing the entropy of the predicted distribution, preventing overconfidence. The experimental results show that focal loss improves calibration without sacrificing accuracy. The study investigates the properties of focal loss, revealing its ability to implicitly maximize entropy and act as a regularization technique. Additionally, models trained with focal loss demonstrate improved calibration and enhanced detec-

tion of out-of-distribution samples compared to temperature scaling. Extensive experiments across computer vision and NLP datasets validate the effectiveness of the proposed approach in achieving state-of-the-art calibration while maintaining high accuracy.

## 3 METHODOLOGY

### 3.1 ARCHITECTURE

As discussed, in this article our objective is to have a network signal its confidence in the prediction it makes. To make this possible, we require the design of a new loss function that includes a confidence measurement. Furthermore, for the new loss function, there is a need to change the architecture of the neural networks. In a typical neural network architecture for classification purposes, the number of classes in the problem domain determines the number of nodes in the last layer. Let X be the feature space, in the case of this paper it would be images, and Y the label space, the distribution of the class labels. The goal of the Neural Networks is to learn a prediction model that has weights, w, that determine its characteristics. We call that function $f_w : X \to Y$. The prediction of the Neural Network would be as follows:

$$\hat{y} = \arg\max_{x \in X} f_w(x) \tag{1}$$

Unlike Ramya Hebbalaguppe et al. Hebbalaguppe et al. (2022) who computed confidence by using the regular neural network architectures, here we want the neural network to signal and learn the confidence itself. In order to make the network returns a confidence score (*confession*), there is a need to add a new node. In this regard, we add a new node to the last layer of the network alongside previously existing nodes. As it is shown in Figure 1, the last node is used for signaling the confession, and other nodes are used for classification purposes as they are in a regular network. The new node (green, $\mathcal{N}_C$, in Figure 1) is being used for the neural network's confidence; the confidence the network has about its prediction is right. With the intention of this new node being consistent with the confidence probability, we used the sigmoid function in the last layer for just the new node, which was used for learning confidence. The other nodes are being used in a way a regular network uses them to learn from data and predict a class.

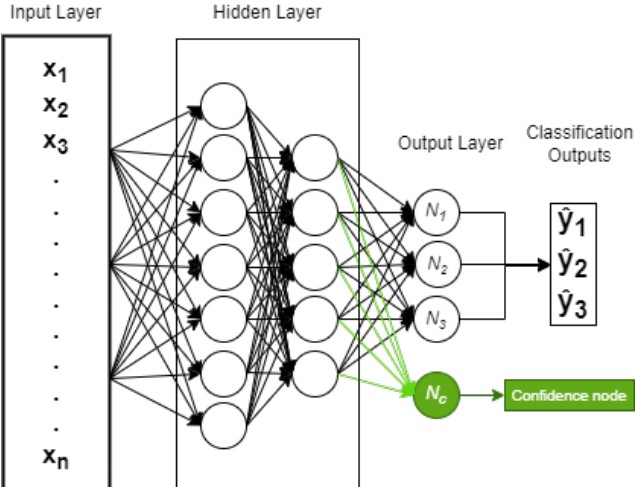

Figure 1: The newly designed architecture of Neural Networks for the classification task

### 3.2 LOSS FUNCTION

With the modified architecture for deploying the confession network, there is a need to design a new loss function to achieve our goal, which is to make NNs return their confidence about the prediction

they have. We require that our networks learn to increase true confidence when the prediction is correct and decrease it when the prediction is false. Therefore, we must distinguish between true and false answers without letting the gradient vanish. To implement this concept, we use the concept of the Margin common to other types of classifiers in machine learning. As we discussed in the background section, one of the ways to increase the accuracy and confidence of true answers for the network is by using a method similar to Margin loss Goldani et al. (2021).

The Margin refers to the difference between the probability of the true class in the network and the maximum probability of other nodes in the network excluding that particular node. Suppose we have a Neural Network that tries to predict class labels with respect to its weights and the input, therefore, $Y = y_i | f_w(x)$. Here the margin will be:

$$\mathcal{M} = \max_{x \in X}(f_w(x)) - \max[f_w(x) - \arg\max_{x \in X} f_w(x)] \tag{2}$$

For instance, suppose we have ten classes in our problem domain, and we have a network for classifying an image belonging to the third class; the margin will be the probability of class three (which is the probability of the third node in the network) subtracting the maximum probability of the nodes in the network excluding the third node. Consequently, if the networks have a correct answer, which means that the maximum probability of nodes in the network is actually for the true class, the margin will be positive, and, it will be negative otherwise. Therefore, by the sign of the margin, we are able to determine whether the network's answer is true or false.

Given a means for distinguishing between true and false predictions, we can revisit the design of the overall loss function. As mentioned above, we want our network to learn to increase the true confidence when the prediction is true and decrease it when the prediction is false. Furthermore, we use Margin to distinguish between true and false answers. If the margin is above zero, it is a true prediction, and it is false otherwise. Accordingly, we use $Max(M, 0)$ for true predictions and $Max(-M, 0)$ for false predictions. Thus, the confession loss function, $\mathcal{L}_\mathcal{C}$, will be:

$$\mathcal{L}_\mathcal{C} = max(\mathcal{M}, 0) \cdot log(P(\mathcal{N}_\mathcal{C})) + max(\mathcal{M}, 0) \cdot log(1 - P(\mathcal{N}_\mathcal{C})) \tag{3}$$

In addition to signaling confidence and having this reflected in the error and training, we also want to predict the true class. To enforce this, we utilize the CCE loss function on predictions made at the final layer. The CCE loss function adds the logarithm of the prediction to our above-mentioned loss function. As a result, the final loss function will be:

$$\mathcal{L} \quad = \quad \sum_{i=1}^{n} (y_i log(P_i) + max(M, 0) \cdot log(P_{n+1}) + max(M, 0) \cdot log(1 - P_{n+1})) \tag{4}$$

In which n is the number of classes in the problem domain, $y_i$ is the $i^{th}$ value in the one-hot vector of the answer and $P_{i+1}$ is probabilities of confession node respectively. It is noteworthy that, since the confession loss, $\mathcal{L}_\mathcal{C}$, is differentiable, we can use other losses instead of CCE Loss which is mentioned above. However, In this paper we mainly focus on just the CCE Loss added to $\mathcal{L}_\mathcal{C}$.

Furthermore, there is a challenge of handling a critical point which is present in the nature of the Confession Loss equation: When $P(\mathcal{N}_\mathcal{C})$ converges to zero, so that, $log(P(\mathcal{N}_\mathcal{C}))$ has an unbound value that prevents training and convergence. In order to prevent this, we bound $P(\mathcal{N}_\mathcal{C})$ between $\epsilon$ and $(1 - \epsilon)$, in which $\epsilon$ is a very small value close to zero, i.e. $(10^-7)$ as we used in our experiments. We, therefore, use Equation 5 to bound the value to be above $\epsilon$ and below $(1 - \epsilon)$.

$$P(\mathcal{N}_\mathcal{C}) = \min(\max(\mathcal{N}_\mathcal{C}, \epsilon), 1 - \epsilon) \tag{5}$$

## 4 EVALUATION AND RESULTS

### 4.1 DATASETS

To evaluate our presented loss, we used the CIFAR-10 and CIFAR-100 to produce a comprehensive evaluation for classification purposes with respect to both the overall accuracy achieved and the value of the confidence node in signaling confidence.

### 4.1.1 CIFAR-10 AND CIFAR-100

The CIFAR-10 dataset contains 32*32 pixels images that have three channels. In the CIFAR-10 dataset, there are 50,000 images for training the networks and 10,000 for testing purposes Krizhevsky et al. (2009). The CIFAR-100 dataset also contains 32*32 pixel images that have three RGB channels, and have the same number of images for training and testing, leading to 600 images for each class in this dataset Krizhevsky et al. (2009). In the training stage, we divide the training images into two sets, a training and a validation set, in such a way that the validation set has 15 percent of the training set images.

## 4.2 NEURAL NETWORKS

For the implementation of the proposed confession networks, we used various architectures which have been shown to have good accuracy for the datasets we consider. We used three different versions of the ResNet networks, ResNet-34, ResNet-50, and ResNet-101. We also tested one alternative in the VGG-16 network. For each of the networks, we trained the original network in the standard manner using Cross Entropy Loss and also adjusted the standard networks by adding a new additional confidence node and using the new loss function. We used Pytorch libraries for creating the networks. It is worth mentioning that for all the networks we used a Stochastic Gradient Descent (SGD) optimizer with an initial learning rate equal to 0.1 and multiplied by 0.1 every three epochs.

## 4.3 RESULTS

### 4.3.1 CIFAR-10

For testing our new loss function and adjusted architecture, first, we tested our networks with the CIFAR-10 dataset. We used Three different ResNet networks (ResNet34, ResNet50, and ResNet101) as described above. Moreover, we used VGG16 to have another type of neural network to test our loss function. All networks have been trained and tested with the exact same hyper-parameters. We used Stochastic Gradient Descent (SGD) for the optimizer with a momentum equal to 0.9 and a learning rate starting from 0.1 and multiplied by 0.5 every three epochs. It is worth mentioning that the loss function we used as a baseline comparison is the Categorical Cross-Entropy (CCE) loss. Therefore, we trained regular networks with CCE and then adjusted the architecture of the networks and subsequently trained them with our new loss function. As is presented in Table 1, we have an improvement in accuracy, Precision, Recall, and adjusted R squared for all the networks. We have a 0.44% increase in the accuracy of Resnet34, 2.87% for Resnet50, 2.16% for Resnet101 and 4.95% for VGG16. It is worth mentioning that the mentioned results are the average of five different runs with the same hyper-parameters and networks.

| Type of Algorithm | Metrics | Resnet 34 | Resnet 50 | Resnet 101 | VGG 16 |
|---|---|---|---|---|---|
| Regular Algorithm | Accuracy | 90.31% | 88.91% | 89.03% | 83.54% |
| | Precision | 90.21% | 88.84% | 89.75% | 83.28% |
| | Recall | 89.97% | 88.51% | 88.56% | 82.66% |
| | Duration of Training (s) | 5848 | **6378** | **8909** | **12588** |
| Our New Loss Function | Accuracy | **91.74%** | **91.78%** | **91.19%** | **88.50%%** |
| | Precision | **90.82%** | **91.35%** | **91.18%** | **87.61%** |
| | Recall | **90.65%** | **90.99%** | **91.05%** | **87.44%%** |
| | Duration of Training (s) | **5567** | 7954 | 9327 | 12826 |

Table 1: Comparison between different networks and algorithms for CIFAR-10

### 4.3.2 CIFAR-100

After testing with the CIFAR-10 dataset, we tested our networks with the CIFAR-100 dataset. We used the previously mentioned networks for this evaluation as well. All the networks have been trained and tested with exact same hyper-parameters as before. We used CCE loss for regular networks and SGD optimizer for both of the loss functions across all the networks. Although the CIFAR-100 dataset is a small dataset that contains just 500 images for each class which leads to

lower accuracy for testing, we tested our new loss function to observe the accuracy and confidence scores for a dataset with a smaller number of examples per class. As is depicted in Table 2, we have an improvement in accuracy, Precision, and Recall for all the networks. We have a 2.64% increase in the accuracy of Resnet34, 4.32% for Resnet50, 3.39% for Resnet101, and 2.24% for VGG16. It is worth mentioning again that the mentioned results are the average of five different runs with the same hyper-parameters and networks.

| Type of Algorithm | Metrics | Resnet 34 | Resnet 50 | Resnet 101 | VGG 16 |
|---|---|---|---|---|---|
| Regular Algorithm | Accuracy | 83.72% | 82.31% | 84.92% | 82.02% |
| | Precision | 70.15% | 70.01% | 72.91% | 72.52% |
| | Recall | 70.42% | 70.11% | 72.83% | 72.76% |
| | Duration of Training (s) | **5576** | **9590** | 13826 | **16239** |
| Our New Loss Function | Accuracy | **86.36%** | **86.63%** | **88.31%** | **84.26%** |
| | Precision | **73.28%** | **73.82%** | **76.19%** | **73.75%** |
| | Recall | **73.67%** | **74.05%** | **76.59%** | **73.97%** |
| | Duration of Training (s) | 7719 | 11409 | **13376** | 17369 |

Table 2: Comparison between different networks and algorithms for Cifar-100

### 4.4 MAKING USE OF THE CONFIDENCE NODE

A natural question that arises at this point is how we can make use of the confidence node. As we described above, we have better accuracy by adjusting the NN and using our new loss function, which considers the confidence node. However, we want to rely on the confidence node to produce trust in our assigned class, alongside the observed increase in accuracy.

To compare the confidence node with regular classification nodes in the networks, we consider the number of positives produced by a given threshold vs. the proportion of true positives. In these plots, we have a true positive rate vs. positive rate for the confidence node (red curve) and the value of the Argmax of regular classification nodes in the networks we tested on.

As shown in Figure 2, which depicts these plots, confidence node curves are above regular nodes; therefore, we have a better true positive rate for the confidence node in every case. The confidence node produces a much higher number of true positives for the same number of positive cases returned across all thresholds. As a result, we can gain better trust from the confidence node in decisions that our network makes, and this will also lead to an increase in the accuracy of the prediction and trust for the network's answer. This also means that for applications where a human is in the loop, for any given threshold for signaling a need for human intervention, the proportion of cases flagged as positive that are correctly classified is much higher.

### 4.5 COMPARISON OF LOSS FUNCTIONS

As depicted in Table 1 and Table 2, in all cases, our adjusted network with the new loss function outperformed regular networks using the CCE loss function. Our loss function increases accuracy while also providing a confidence measure that can be used to gauge the validity of our classification.

## 5 CONCLUSION

In this paper, we proposed a novel method for signaling the confidence of a neural network's prediction through the use of the added node called the confidence node. Although there are some existing statistical approaches to measuring a neural network's confidence, in this paper, we design a new loss function to predict confidence directly from the network. With the aim of doing so, we modify the architecture of neural networks and propose an associated loss function. This produces an improvement in accuracy in classification tasks for the CIFAR-10 and CIFAR-100 datasets by testing different networks (ResNet 34, 50, 101, and VGG 16), alongside producing valuable confidence measures directly from the Neural Network, to get a sense of how we could be sure about the prediction of the Neural Network.

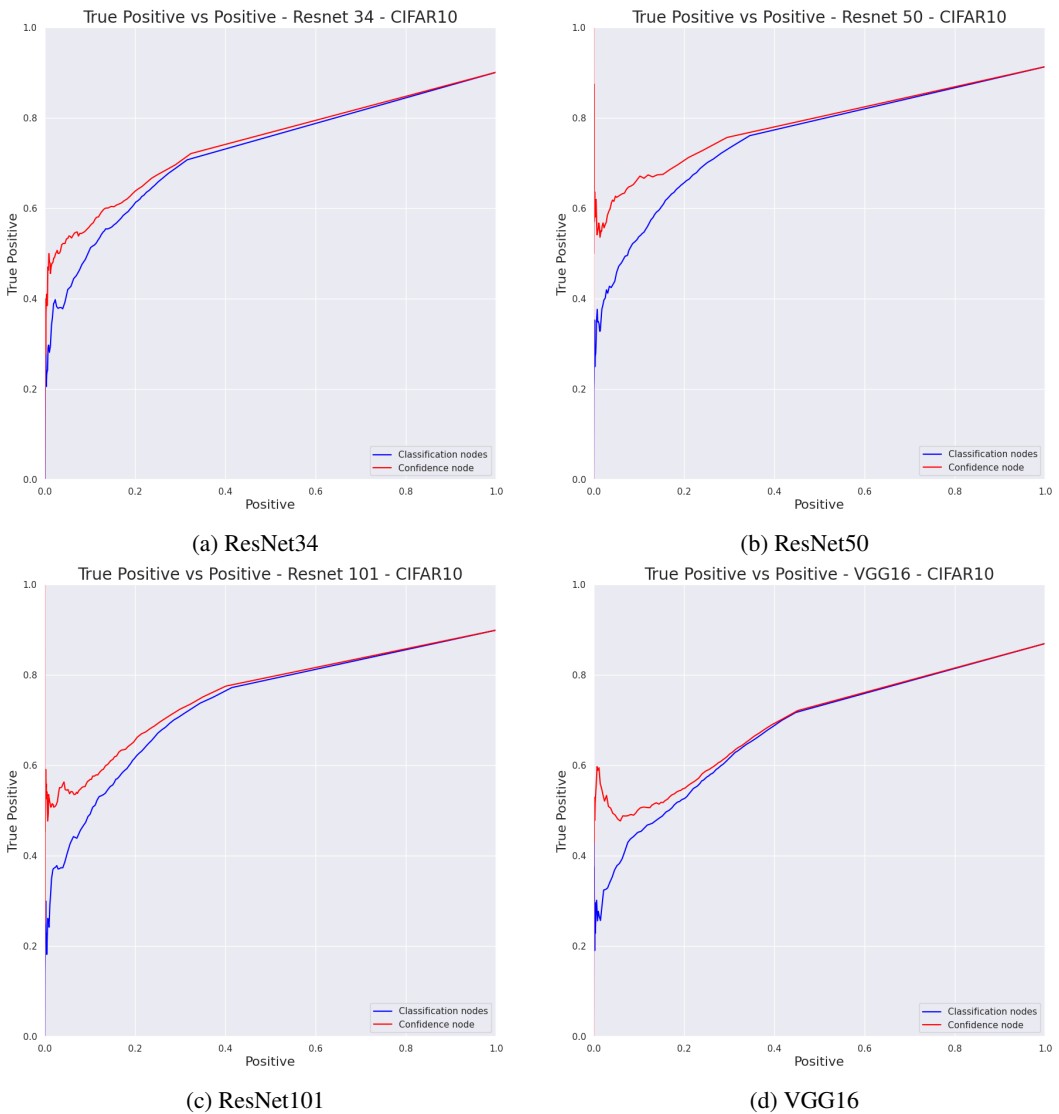

Figure 2: True positive rate vs. Positive rate for Cifar-10 dataset

We showed that by using the confidence node, we would have a higher proportion of true positives than just using the classification node alone. There are many applications where the confidence of neural networks is critical. For instance, this could be used in autonomous driving to make sure that our predictions relating to the surrounding area of the vehicle are true, or in important medical domains that possibly include a human in the loop.

# 6   FUTURE WORK

In this paper, we used ResNet-34, ResNet-50, ResBet-101 and VGG16 to test our proposed neural network architecture and new loss function. Further extensions of this work may involve examination of a greater variety of different neural networks to observe the degree of gain in performance.

We also intend to consider changing the position of confession nodes from the last layer to some other layer to observe the difference arising from varying the position of these nodes. Having loss functions other than CCE could also be an interesting test towards benchmarking this new loss function from an additional perspective.

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
