# OpenReview forum: "Confession Networks: Boosting Accuracy and Improving Confidence in Classification"
_ICLR.cc/2024/Conference — Submitted to ICLR 2024_

### Official Review · Reviewer_dbJp · 2023-10-20

**Soundness:** 2 fair
**Presentation:** 2 fair
**Contribution:** 2 fair
**Rating:** 3
**Confidence:** 5

**Summary:**

This paper proposes adding one new node to the last layer of image classification models so that the new node can indicate the correctness of classifications. The proposed approach is also meant to improve the accuracy of the models. Some experiments are presented on the CIFAR datasets.

**Strengths:**

The goals that the paper pursues are clearly defined: boosting accuracy, faster training, estimating the confidence of outputs, …

The method itself is quite simple, adding a single node in the last fully connected layer of the network. This simplicity could be the strength of the method, if the method can achieve any of its goals/claims on standard models with high accuracy.

**Weaknesses:**

The main weakness, in my view, is the reported results in the paper.

The accuracy of the original models that authors have trained (with and without their proposed method) is surprisingly low. For example, the accuracy of the ResNet-34 that authors have trained on CIFAR-10 is about 90.3% while the accuracy of pretrained models on Hugging Face and PyTorch libraries are between 95 to 98%. This indicates that authors are not using the best training methods available in the literature for ResNet and other models.

Ultimately, the paper’s proposed loss function improves the accuracy of 90.3% to 91.7% which is still much lower than 95-98%. The accuracy of the ViT model trained on CIFAR-10 is already 98%. Why would anyone leave the pretrained model with accuracy of 98%, and use the authors’ method to ultimately achieve an accuracy of 91.7%?

So, with the current experiments, there does not seem to be any advantage in using this paper’s method. The training speed with the proposed loss function also does not show a significant improvement. This would change if authors demonstrate that their method can improve the accuracy and confidence of standard models with high accuracy.

Note that the paper does not state that it is compromising accuracy for confidence. The claim in the paper is that it is boosting the accuracy of models as well as improving the confidence.

------------

It is not clear to me what the horizontal axis represents in the plots in Figure 2. In the description of Figure 2, a threshold is mentioned, but it is not clear to me what that threshold is and how it is reflected in the plots.

It seems to me that Figure 2 compared the confidence node with the top softmax score of the same models. I think the comparison could be made with the softmax score of the original models.

In the context of Figure 2, authors only mention the true-positive rate. But that does not give the whole picture. One would also need to know the false negative rates, etc, for all the scenarios.

-----------

Experiments are limited to CIFAR-10 and CIFAR-100 datasets. This makes the results thin and not convincing. Experimenting on Imagenet has been a standard for a few years.

While the paper mentions at the beginning that “One motivation for this work is in decision-critical settings”, eventually its experiments are on CIFAR datasets.

------------

In my view, there is no need to dedicate a paragraph to describe the CIFAR datasets. The audience of ICLR is well familiar with these datasets. It is also common practice to have the details about the models and datasets in the appendix.

**Questions:**

Please see questions under weaknesses.

Do authors think they can extend their experiments to Imagenet?

Can authors perform their experiments on models with +95% accuracy on the CIFAR-10 dataset?

---

### Official Review · Reviewer_nRM1 · 2023-10-28

**Soundness:** 2 fair
**Presentation:** 2 fair
**Contribution:** 2 fair
**Rating:** 5
**Confidence:** 3

**Summary:**

This paper introduces a new method for neural networks to express their confidence in classification predictions using a novel loss function. The main goals are to design a loss function that provides a confidence measure alongside predictions and to assess its impact on network performance.

**Strengths:**

This paper raises a practical issue concerning neural networks, particularly when they are applied in domain-specific contexts. In many cases, the desired outcome goes beyond standard accuracy metrics. An adaptive framework that allows neural networks to optimise for multiple objectives becomes crucial in such scenarios. Exploring the idea of providing a level of confidence for predictions is indeed a valuable area to investigate.

**Weaknesses:**

While it's a promising area for exploration, I noticed that the analysis of the newly introduced element (e.g., the confidence node) is somewhat lacking in depth. The authors might also consider comparing their approach with alternative strategies that could achieve similar results.

**Questions:**

What measures can be taken to avoid overfitting of the confidence node during the training stage?

---

### Official Review · Reviewer_8iyk · 2023-10-29

**Soundness:** 1 poor
**Presentation:** 2 fair
**Contribution:** 2 fair
**Rating:** 1
**Confidence:** 5

**Summary:**

The paper proposes to predict the confidence of a neural network by adding a confidence node and changing the loss function a little bit.

**Strengths:**

Addresses an important problem, since neural networks are known to be very confident on data far away from the training examples.

**Weaknesses:**

- The method is not described well, since the description has some problems. For example eq (1), which is supposed to describe the prediction of a neural network, is wrong since argmax over x in X belongs to X, so y hat belongs to X.
- A confidence node will not work because because a MLP (multi-layer perceptron) with ReLU activation is actually a piecewise linear function, so the confidence output will be piecewise linear on the domain X. Some of the regions of linearity will be bounded but some will be infinite. On the infinite regions the output will take some very large values in some regions of the space X, far away from the training examples, so the confidence will be very high even though it should not be. This is independent of the loss function because through a loss function one will never be able to make the function constant on all the infinite regions.
- Experiments are seriously lacking. First, experiments on CIFAR-10 and CIFAR-100 are not enough, experiments on other datasets are needed. Second, to show that the proposed confidence works, one should test it on data far away from the training examples, where it matters. This the problem of OOD (out of distribution) detection, which was slightly touched in the literature review. However, the authors fail to compare their method with any of the OOD detection algorithms from the literature.

**Questions:**

- How does the proposed algorithm compare with XU et al 2022 for OOD detection with CIFAR-10 as in distribution (ID) and CIFAR-100, Imagenet or SVHN as OOD data? Similarly with CIFAR-100 as ID and CIFAR-10, Imagenet and SVHN as OOD? How does it compare with other recent OOD detection method on the same datasets?
- How does the method scale to Imagenet as ID data and Food Network, iNaturalist as OOD data? See "Openood: Benchmarking generalized out-of-distribution detection" from NeurIPS 2022 for details.

---

### Official Review · Reviewer_TtcN · 2023-10-31

**Soundness:** 3 good
**Presentation:** 2 fair
**Contribution:** 2 fair
**Rating:** 5
**Confidence:** 4

**Summary:**

This manuscript presents an approach to estimate sample-based confidence score of prediction for multi-class classification setting. Method is based on max margin calibration, adapted to neural network loss function. Experimental evaluation is done using CIFAR-10 and CIFAR-100 image dataset outperforming model trained with standard loss function (considering classification accuracy and calibrated uncertainty).

**Strengths:**

Paper proposes a simple yet effective margin loss based classifier confidence calibration. It is easy to apply for existing classification models (but needs full training process). Promising results in image recognition domain is achieved compared to standard loss function and training, improving the accuracy and confidence quantification. There are some limitations in the experiments (see weaknesses).

**Weaknesses:**

The background and literature reviews lacks some of the related previous approaches of uncertainty and confidence quantification (e.g., Bayesian NNs, post-processing calibration methods, and out-of-distribution (OOD) type estimation). Although, experiments shows promising results, they are quite limited. There should be more detailed and comprehensive evaluation considering different DNN models and variety of datasets from different domain to support the findings and proposed approach better. Also, there is no any comparison to previous confidence calibration approaches.

**Questions:**

- How the proposed approach compares to previous approaches theoretically and experimentally? E.g., different post-processing methods, or ones in OOD literature (see e.g., [1] for related approach)
- What is causing the peak with the small positive rate in Figure 2, when utilising the confidence score?

[1] DeVries et al. (2018) Learning Confidence for Out-of-Distribution Detection in Neural Networks, arXiv:1802.04865.

---

### Meta-Review · Area_Chair_dR1y · 2023-12-18

**Metareview:**

The reviewers make strong points against the paper, with several key weaknesses mentioned (8iyk, dbJp) on description and evaluation. The authors have made no reply to the reviews. It is recommended to follow the reviewers recommendations to improve the paper's content.

**Justification For Why Not Higher Score:**

Very low scores, strong arguments against, no rebuttal.

**Justification For Why Not Lower Score:**

N/A

---

### Decision · Program_Chairs · 2024-01-16

Reject